# ChartAB: A Benchmark for Chart Grounding & Dense Alignment

## Abstract

Charts play an important role in visualization, reasoning, data analysis, and the exchange of ideas among humans. However, existing vision-language models (VLMs) still lack accurate perception of details and struggle to extract fine-grained structures from charts. Such limitations in chart grounding also hinder their ability to compare multiple charts and reason over them. In this paper, we introduce a novel "**ChartA**lign **B**enchmark (ChartAB)" to provide a comprehensive evaluation of VLMs in chart grounding tasks, i.e., extracting tabular data, localizing visualization elements, and recognizing various attributes from charts of diverse types and complexities. We design a JSON template to facilitate the calculation of evaluation metrics specifically tailored for each grounding task. By incorporating a novel two-stage inference workflow, the benchmark can further evaluate VLMs capability to align and compare elements/attributes across two charts. Our analysis of evaluations on several recent VLMs reveals new insights into their perception biases, weaknesses, robustness, and hallucinations in chart understanding. These findings highlight the fine-grained discrepancies among VLMs in chart understanding tasks and point to specific skills that need to be strengthened in current models.

## 1 Introduction

Recent large multimodal models (LMMs), such as vision-language models (VLMs), have achieved remarkable breakthroughs in aligning the visual modality with language models, enabling challenging language-level reasoning on visual input signals and opening the door to a wide range of applications that naturally rely on interactions between the two modalities (Alayrac et al., 2022; Li et al., 2023; Liu et al., 2023b). One critical class of applications is chart understanding and reasoning, which has broad use in finance, data science, mass media, biology, and other scientific domains where ideas and information are communicated through visualizations. In these applications, measuring numerical values in charts, comparing visual elements (e.g., bars or curves), mapping correspondences between colors, numbers, names, or markers, and recognizing attributes are essential skills for downstream tasks. Most of these tasks require accurate grounding of the structured details in charts. Moreover, dense alignment of elements across multiple charts is also a widely needed skill in practical scenarios. These challenges present new open problems for VLMs.

Instead of focusing on charts, existing VLMs have primarily been pretrained and finetuned on natural images and common questions/instructions, which are not fully compatible with chart understanding tasks (Yao et al., 2024; Laurençon et al., 2024). Unlike perceiving objects shapes, poses, and semantic meanings in natural images, accurate measurement and comparison of geometric/graphic components, understanding of their structure and layout, and manipulation of their positions and rich textual content are more critical for perception and reasoning with chart images. However, it remains challenging for VLMs to acquire these capabilities, often leading to hallucinations and misinterpretations in chart-centric tasks (Masry et al., 2022; Xia et al., 2024).

Despite the recent growing interest in chart-related tasks, existing VLMs and benchmarks specifically designed for charts usually focus on simple QA tasks (Masry et al., 2022; 2025; Wang et al., 2024b; Li & Tajbakhsh, 2023), which cannot comprehensively assess the capabilities of VLMs in grounding and understanding chart components for more general-purpose tasks. Moreover, the alignment of

layouts and components across multiple charts has not been explored in previous work. Hence, there remains a lack of benchmarks dedicated to evaluating these critical skills.

In this paper, we take the first step toward systematically evaluating and analyzing general-purpose VLMs on chart grounding and multi-chart dense alignment. We formally categorize the information to be grounded in a chart into two dimensions: (1) **data**, and (2) **attributes** (e.g., colors, styles, legends, sizes, positions) that define the visualization design, components, and layout. We define the *chart grounding task* as extracting both the underlying data table and the associated attributes from a chart image, and the *dense alignment task* as identifying correspondences and differences between two charts. Together, these tasks represent fundamental capabilities and critical subroutines required for a wide range of chart-centric applications.

To this end, we develop a comprehensive benchmark using pairs of similar charts to evaluate model performance on the two tasks with respect to each type of information in the two categories. To create a pair of similar charts, we perturb an existing chart by randomly modifying (1) one or a few data cells in the data table and/or (2) an attribute in the script used to generate the original chart. To maximize the potential of VLMs and evaluate their full capabilities, we propose a multi-stage information extraction and query pipeline. In this pipeline, VLMs are first queried with a grounding task targeting specified information in each chart, followed by a comparison of the grounding results between the two charts. The pipeline leverages structured JSON templates to guide the grounding and alignment of different types of information. In addition, we introduce several novel evaluation metrics that account for the symmetry and ambiguity inherent in various types of information, thereby enabling more reliable quantitative comparisons across different VLMs.

Our analysis reveals the weaknesses of existing VLMs in chart perception and understanding for dense grounding and alignment. The observed errors highlight their biases and hallucinations regarding certain chart components, offering critical insights for improving VLMs. The evaluation results further show how differences across models, chart types, and queried data/attributes influence benchmarking performance. In addition, we assess the robustness of VLMs in data grounding and alignment under different attribute variations, such as changes in chart type or color schemes.

**Our contributions and novelties** are summarized as follows:

- We introduce the first comprehensive benchmark, "ChartAB" to systematically evaluate VLMs' capabilities in dense grounding and multi-chart alignment of data and attributes in chart images.

- We propose a holistic evaluation suite, including a multi-stage pipeline converting charts into JSON files with specific templates for tasks regarding data/attributes, and a rating scheme of the grounding/alignment performance based on VLMs' answers.

- Our evaluation and analysis of existing VLMs reveal weaknesses in fine-grained chart understanding, highlight hallucinations, and expose biases in their vision encoders when perceiving critical chart features and structures.

- We evaluate the robustness of grounding and alignment in data under perturbations of attributes. It provides novel insights for the design of high-quality charts.

## 2 RELATED WORK

**VLMs for Charts.** Vision-language models have shown significant advancements in chart understanding tasks. They can be broadly classified into (1) general-purpose multimodal models and (2) chart-specialized models. General-purpose models include proprietary ones (Hurst et al., 2024) and open-source ones (Abdin et al., 2024; Chen et al., 2024; Liu et al., 2023a; Bai et al., 2025). Chart-specialized models (Zhang et al., 2024b; Masry et al., 2024; Xia et al., 2024; Meng et al., 2024) demonstrate strong performance on chart benchmarks; however, they are limited by instruction tuning on specific tasks, which restricts dense-level understanding, and are further hindered by incompatible pipelines that often rely on predefined routines to handle task requirements.

**Chart Understanding Benchmarks.** Current chart benchmarks evaluate VLMs on specific tasks including question answering (Methani et al., 2020; Masry et al., 2022), summarization (Kantharaj et al., 2022b), explanation-generation (Kantharaj et al., 2022a). Multi-task benchmarks including ChartLlama Han et al. (2023), ChartX Xia et al. (2024) perform agglomeration of various modalities

(like chart data, description, summary) for the downstream tasks. Recent works specifically focus on expanding QA scope to overcome increased saturation by VLMs, for example CharXiv Wang et al. (2024b) focuses on charts in research papers, SciGraphQA Li & Tajbakhsh (2023) evaluates multi-turn QA, MultiChartQA Zhu et al. (2024) evaluates multi-hop reasoning on multiple related charts, ChartQAPro Masry et al. (2025) includes diverse visualizations such as dashboards, infographs, and flexible questions (hypothetical, unanswerable).

**Visual Grounding.** The dense-level understanding abilities of VLMs have been extensively enhanced through visual grounding. DePlot Liu et al. (2022) trained a transformer for image-to-CSV generation, introducing a novel table comparison method for evaluation. StructChart Xia et al. (2023) proposed module-based augmentation for efficient grounding of chart data and plot code in downstream applications. Beyond charts, the Grounded-SAM model (Ren et al., 2024) leverages Grounding-DINO (Liu et al., 2024) for improved dense-level open-set object tracking. BLIP-2 Li et al. (2023) has been widely integrated into VLMs for VQA-related tasks. LLaVA-Grounded Zhang et al. (2024a) enables detailed text descriptions of multi-object natural images by leveraging imagetext grounding for instruction tuning.

**Multi-Image Reasoning.** Multiple benchmarks have been developed to evaluate VLMs on multi-image reasoning. MMMU Yue et al. (2024) includes interleaved examples with multiple images from medical, cartoon, art, and technical domains. MUIRBench Wang et al. (2024a) focuses on multi-chart diagram QA but is limited to coarse-level understanding. MMIR Zhao et al. (2024) addresses chart understanding through cross-modal alignment, i.e., plotting-code correctness relative to the chart image. MileBench Song et al. (2024) introduces semantic understanding tasks involving text-rich images, emphasizing text extraction and comprehension in OCR, documents, and slides.

## 3 CHARTAB: CHART GROUNDING AND ALIGNMENT BENCHMARK

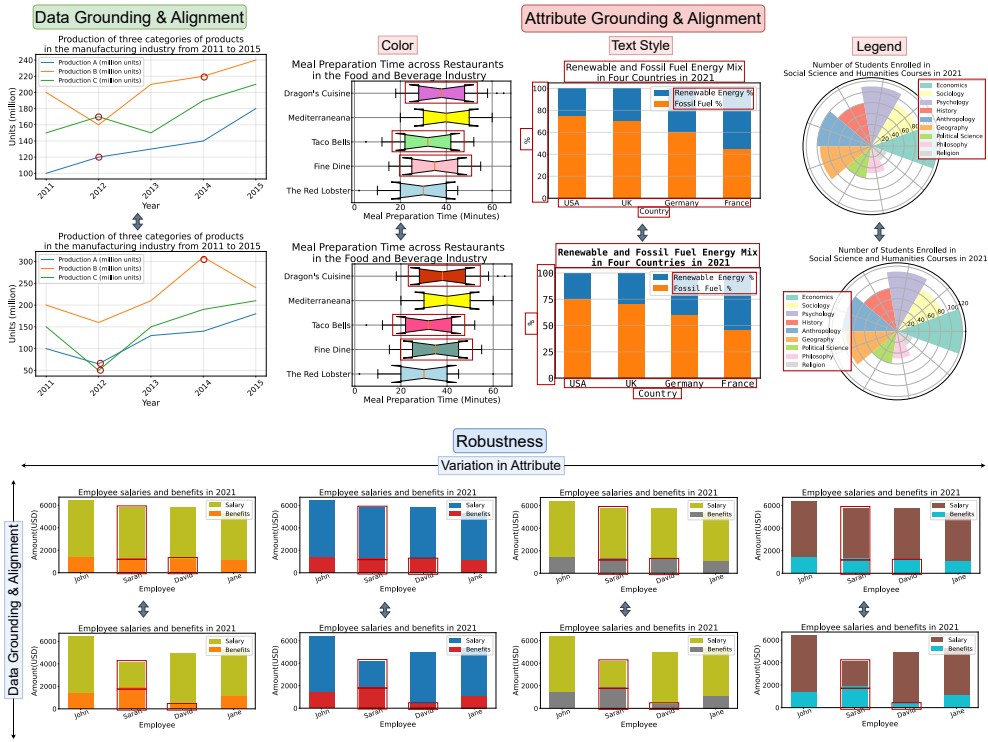

Figure 1: **Example pairs from ChartAB.** ChartAB evaluates dense-level chart understanding in VLMs through chart grounding and dense alignment across pairs of chart images. Pairs in the *Data Grounding & Alignment* task differ in the underlying data values visualized by the charts. Pairs in the *Attribute Grounding & Alignment* task differ in attributes, i.e., visual appearance such as color, legend position, or text style. The *Robustness* task contains multiple pairs that share identical data differences while varying in attributes (e.g., color, as illustrated).

We introduce `ChartAB`, the first benchmark designed to evaluate vision-language models (VLMs) on dense level chart understanding. The benchmark focuses on two core capabilities essential to chart reasoning: (1) *grounding*: extracting structured information from a single chart image, and (2) *alignment*: identifying fine-grained differences between a pair of similar charts. These capabilities serve as critical building blocks for a wide range of downstream applications. We develop a novel two-stage pipeline for performing an integrated evaluation of a VLM for these capabilities.

We construct the `ChartAB` dataset for evaluating these capabilities. We include several examples from our dataset in Figure 1. The dataset encompasses diverse chart types from various topics along with ground-truth labels with extremely high level of precision. By isolating and rigorously evaluating *grounding* and *alignment* abilities, `ChartAB` offers a deeper diagnostic understanding of VLMs for their perceptual accuracy, reasoning limits, and alignment behavior in structured visual domains.

## 3.1 `ChartAB` DATASET CONSTRUCTION

We use the ChartX dataset Xia et al. (2024) as the source dataset. It encompasses diverse chart types from various domains, including commerce, industry, lifestyle, society, and culture, and provides both csv data and plotting code for each chart.

To create pairs of similar chart images, we start with an image from the ChartX dataset and apply controlled perturbations to the plotting code, followed by executing the code to render the chart images. We leverage each charts source data and code (i.e., CSV table and plotting script) to generate precise ground-truth values.

We selected nine diverse chart types to perform data and attribute perturbations: (1) *simple charts*: bar chart, bar-numbered chart, line chart, and line-numbered chart; (2) *complex charts* 3D chart, box chart, radar chart, rose chart, and multi-axes chart. More details on dataset construction are provided in A.3.

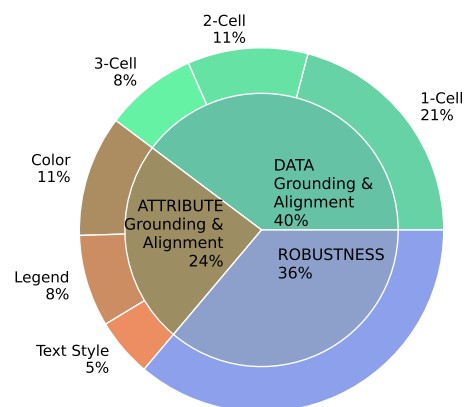

Figure 2: **Dataset statistics for `ChartAB`.** The benchmark includes 9k pairs of chart images. *Data Grounding & Alignment* pairs differ in one to three data cells. *Attribute Grounding & Alignment* pairs differ in color, legend position, or text style. *Robustness* tasks include multiple pairs that share identical data differences but differ in attributes.

## 3.2 GROUNDING OF SINGLE CHART

Dense level understanding requires the extraction of precise semantic information from chart images, including data, and attributes. General-purpose VLMs face challenges in performing it due to their reliance on global visual embeddings, which fail to represent a chart image's object-level details Xu et al. (2023), which are critical in chart reasoning. Prior works primarily use QA on the chart image, which tends to obscure semantic understanding and exacerbate VLM's cross-modal inconsistencies Huang et al. (2024). To enable more interpretable and compositional reasoning, we first analyze the model's ability to ground the chart information (i.e. data & attributes) to textual form.

We formalize *grounding* task: a chart image as input, resulting in a structured textual representation of its contents. It is performed over two key semantic layers (1) *data*: underlying data table that the chart visualizes. We prompt the model to generate a standard csv-style representation of the data table capturing the headers (i.e. rows and columns) and cell values. (2) *visual attributes*: visual attributes encompasses various constituents impacting visual appearance, such as color mappings, legend positions, and text styles. We define attribute-specific JSON-style templates for each of them, prompting the model to generate their structured representations. Grounding the chart image into textual form isolates the model's perceptual ability from downstream prompt variation or instruction complexity. This helps in building a foundation for the subsequent dense alignment tasks, while also enabling failure analysis of VLM in perceiving chart components.

### 3.3 DENSE ALIGNMENT BETWEEN TWO CHARTS

While single chart grounding evaluates a models perception in isolation, real-world use cases often require comparing similar charts to detect subtle differences among the charts. To simulate this, we define a dense alignment task where the model must identify fine-grained discrepancies across chart pairs. Crucially, this task builds on grounded representations, allowing us to isolate and evaluate comparative reasoning for given chart-pairs. As shown in our ablation studies A.7.2, direct alignment without grounding yields significantly weaker performance, highlighting the necessity of grounding for subsequent dense-level alignment.

We define dense alignment as a comparison between two chart images that diverge in finer-level chart details. It constitutes 3 tasks: *Data Alignment*, *Attribute Alignment*, *Robustness*. The 3 alignment tasks consist of ∼ 3,600, ∼ 2,000, ∼ 3,300 instances respectively. For data alignment & attribute alignment, each instance consists of pair of chart-images differing in underlying data & attribute ( color/legend/text-style) respectively. For Robustness, each instance contains 5 pairs of chart-images, each pair with identical difference in data but variation in attribute (e.g. color of bars) across the 5 pairs. Each alignment task challenges the model to identify the set of divergent elements, and produce a structured JSON capturing these differences, enabling evaluation at the element level. The task details are discussed in A.4.

### 3.4 A TWO-STAGE EVALUATION PIPELINE

The two-stage approach fundamentally envisions the dense-alignment task as decomposable into sub-tasks utilizing the visual-to-text grounding to perform finer-level analysis. The task decomposition enables splitting complex finer-level reasoning into smaller steps for efficient element-wise comparisons and handling model biases.

It is inspired from the multi-step approach used in SOTA reasoning models. Figure 3 shows color-alignment evaluation for the o4-mini model OpenAI (2025). The model's reasoning window shows grounding of box (i.e. visual encoding) colors from each of two box plots (i.e. charts) respectively, followed by dense-alignment on the grounded color information. This multi-step approach of the model validates our task decomposition approach and its ability for efficient multi-image dense alignment.

We perform zero-shot inference using natural-language instructions combined with JSON template as output format. This enables easy instruction following and flexible output parsing and evaluation. As shown in Figure 4 *First-stage* results in an intermediate grounding with semantic information on underlying data. The interpretable nature and element wise representation enables subsequent reasoning for fine-grained alignment. *Second-stage* involves VLM reasoning by applying discriminative comparison on the grounded results from first stage to perform the specific dense alignment task resulting in final JSON output.

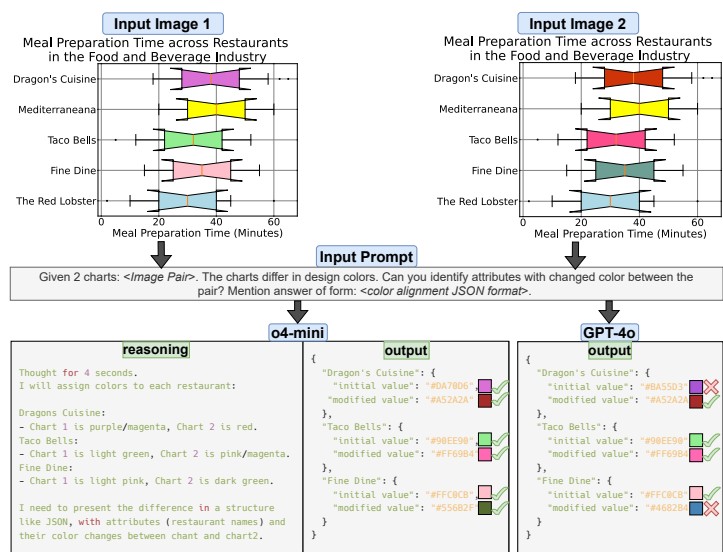

Figure 3: **Multi-step color alignment with o4-mini.** The o4-mini model decomposes the alignment task, using the reasoning window to first ground the colors of each chart before performing alignment, yielding a more accurate result than GPT-4o, which performs alignment directly without intermediate grounding.

The second stage is critical for evaluating end-to-end alignment by requiring VLMs to perform semantic comparison over grounded outputs, beyond just surface-level extraction. It also mitigates

grounding ambiguities and enables additional contextual information thus offering a more human-like assessment of alignment ability. Pipeline details are discussed in A.5.

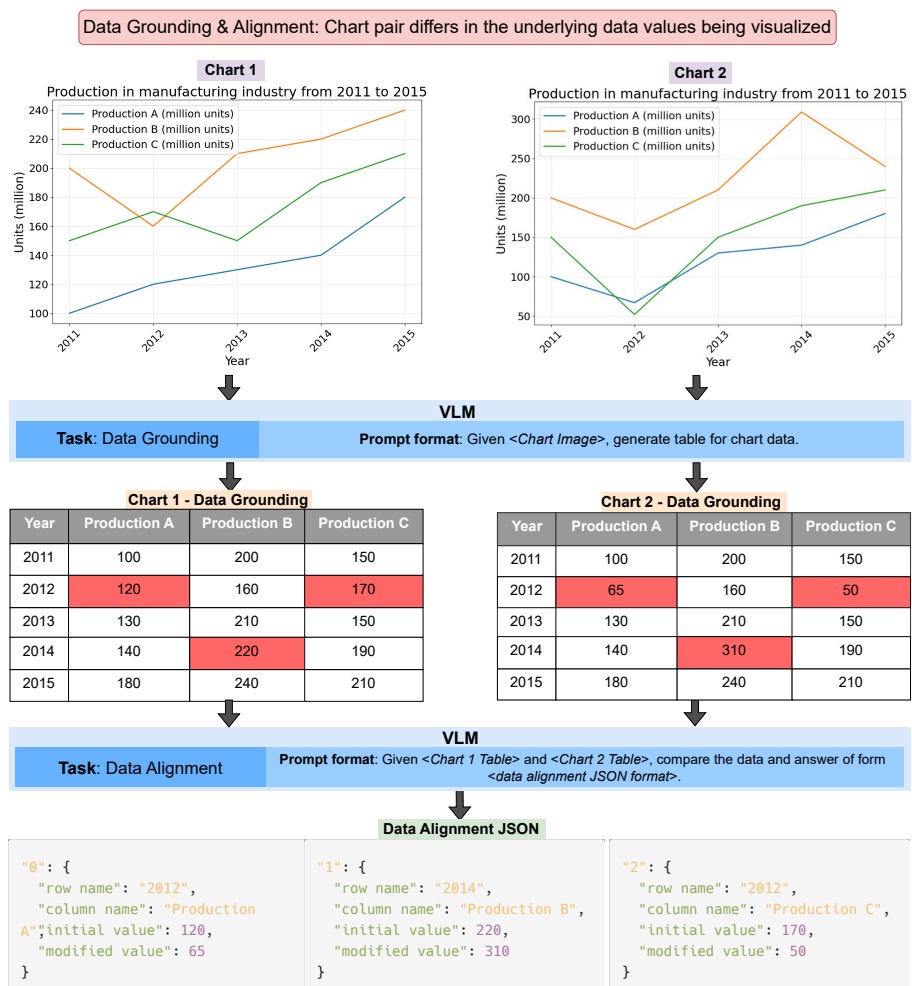

Figure 4: **Two-Stage Evaluation Pipeline of dense level understanding of data in `ChartAB`.** The first stage focuses on grounding the data in each chart to a table, while the second stage i.e. alignment requires the VLMs to find the difference between the two charts' tables and output a JSON file listing the different cells in the two tables. Evaluation of other categories adopts similar multi-stage pipelines, with details in the Appendix (Figures 15, 16, 17).

## 3.5 DOWNSTREAM QA EVALUATION

The practical application of VLM on charts is towards diverse downstream tasks requiring complex reasoning. The grounding & alignment ability serve as building blocks for the downstream reasoning hence form cornerstone for VLM's effectiveness on the subsequent corresponding downstream tasks. And errors in grounding/alignment tasks are common reasons behind failure on these high-level reasoning tasks. For analyzing downstream abilities, we evaluate VLM for Question-Answering (QA) on the chart image. The QA evaluation as most widely applied downstream task (discussed in 2) along with precise objective scoring motivate its selection. ChartX dataset's Xia et al. (2024) QA set is utilized. The QA set's questions are generated focusing on chart data with 1-word answer format with binary result which can be answered using knowledge of csv data table.

## 3.6 EVALUATION METRIC

For evaluating *alignment* of chart-pair, the outputs structured as JSON are utilized to capture differences across $\mathcal{N}$ constituents involved in the task (e.g. colors of bars/lines, altered data points,

or text regions). Each constituent $i$ is assessed for its alignment accuracy $acc_i$, and the **combined accuracy** is averaged and **normalized** to [0 - 10] resulting in alignment score $\mathcal{S}_{\text{align}}$. This enables us to effectively differentiate model performance across tasks, and quantify performance aspects for visualized data and visual attributes as part of dense alignment. The alignment score $\mathcal{S}_{\text{align}} = 10 \cdot \left( \frac{1}{N} \sum i = 1^N \text{acc}_i(\text{chart-pair}) \right)$. Task-wise metrics are discussed in A.6

We also evaluate *grounding* of chart image by calculating the correctness of semantic elements (e.g. color of encoding like bar, legend position, size of text) corresponding to a chart image. For (1) *legend* grounding (analyzed in Figure 8) & (2) text-style grounding (analyzed in Figure 6) we simply apply categorical correctness. (3) Color grounding (analyzed in Figure 7) is evaluated using L2 distance in the RGB color space.

## 4 EXPERIMENTS & ANALYSIS

We evaluate 4 open-source VLM families: Phi-3.5 vision-instruct Abdin et al. (2024), InternVL-2.5 Chen et al. (2024), LLaVA-1.6 Liu et al. (2023a), QWEN-2.5 VL Bai et al. (2025). And GPT-4o Hurst et al. (2024) as proprietary model. We also evaluate chart-specialized VLMs, including TinyChart Zhang et al. (2024b) & ChartGemma Masry et al. (2024). However, due to their task-specific training (discussed in 2), these models show a collapse of instruction following capabilities and fail to output the required JSON format needed for evaluation. Further discussed in A.7.1. Ablations found in A.7.2.

**Downstream QA Evaluation.** The QA evaluation pipeline involves grounding of the chart image into csv (table) in the first stage followed by answering of question using the grounding table as input. With an aim to analyze impact of fundamental grounding/alignment on downstream QA task, we correlate chart-wise QA vs data Alignment performance.

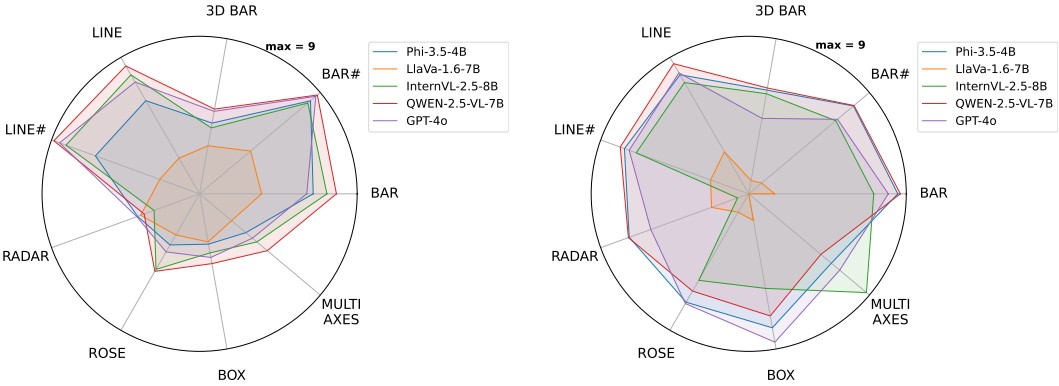

Figure 5: (a) Comparing VLMs on **Data alignment** tasks when two charts' data tables differ in only **one cell**. Llava-1.6 is worse than most other VLMs. QWEN-2.5-VL outperforms GPT-4o on most chart types. Related discussion in Finding 1. (b) **Color alignment** between two charts on fine-grained visual elements (e.g., bars, lines, sectors). VLMs perform better on simpler and more common charts. Related discussion in Finding 1.

> **Finding 1**
>
> VLMs' dense grounding and alignment of data/color information are not satisfying on complex charts.

Compared to simpler and more common charts, e.g., bar/line charts and numbered bar/line charts, dense grounding/alignment on complex charts such as 3D/box/radar/rose/multi-axes charts with more components and irregular layouts is more challenging to most VLMs. Despite the similar alignment performance for *legend* (Figure 12a) and *text-style* (Figure 12b) between simple vs. complex charts, the *color* and *data* alignment (Figure 5) on complex charts are much poorer than those on simple charts. The color grounding requires identifying each constituent's visual encoding and corresponding color, while the visualized-data grounding needs to find the mapping from visual encoding to numeric

values. Hence, complex layouts with more components make these tasks more difficult. In contrast, identifying the position of legends and text styles (which both have limited options) is easier and less affected by the chart complexity.

**Finding 2**

VLMs' text-style grounding and alignment performance is poor in general, and it varies across text size, weight, and font family.

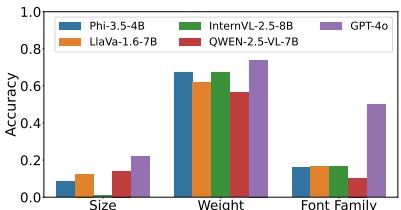

Figure 6 shows that most VLMs fail to detect the correct text size and font family, suffering from an accuracy below 20% (except GPT-4o's performance on font family grounding). These indicate a lack of knowledge on these two text attributes. VLMs' performance on text weight ((light/normal/bold)) is much better (∼60%) and close to each other, but still not satisfying. Although LLMs can select reasonable text sizes in code generation for plots, they tend to rely on the default sizes in their priors or relative sizes to other chart components. They still lack sufficient capability to identify text sizes in chart images.

Figure 6: **Text-style grounding** on size, weight, and font family. Accuracy is low across most VLMs, highlighting lack of style knowledge (Finding 4).

**Finding 3**

VLMs' weak color recognition ability.

As shown in Figure 7, all models' color grounding error (L2 distance in RGB space) has a median exceeding 50. This suggests their inability to understand color shades beyond common ones, e.g., red, blue, green, etc., which exposes their weaknesses in color recognition.

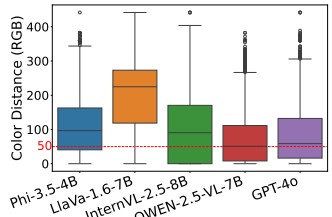

The lack of color understanding affects the perception of detailed differences in charts and leads to mismatches in color-related/conditioned reasoning tasks. Consequently, the VLMs' performance in color alignment tasks (Figure 5) is consistent with that on color grounding. These results suggest improving the color understanding capability by adding more color-sensitive data to VLM training.

Figure 7: **Color recognition** measured by L2 errors in RGB space. Median errors exceed 50 for all VLMs, indicating weak color recognition (Finding 3).

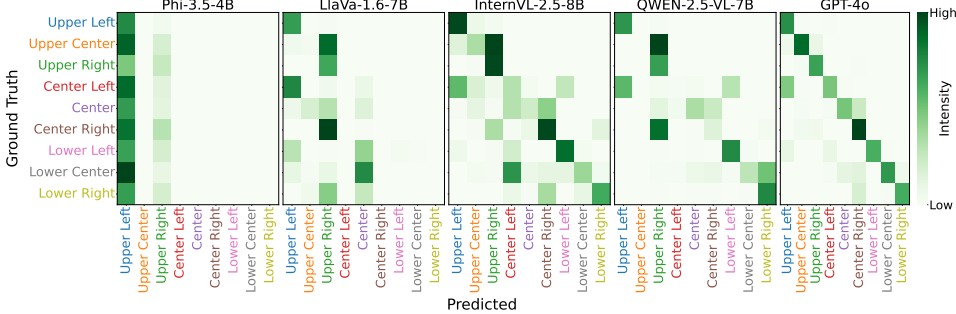

Figure 8: **Confusion matrix of legend position grounding for each VLM.** The dark non-diagonal entries highlight the fail patterns and biases of incorrectly identifying position-$i$ as position-$j$. Phi-3.5 exhibits a severe bias towards *upper-left* position while GPT-4o shows the minimal bias. More discussion is provided below Finding 2.

**Finding 4**

Most VLMs suffer from biases when allocating the position of legends.

The grounding of the legend's position (Figure 8) suffers from a strong bias of pretrained VLMs. The Phi-3.5 model shows the strongest prior towards the *upper-left* position. The 7-8B scale VLMs, e.g., LlaVa-1.6, Inten-VL-2.5, QWEN-2.5-VL, all show a similar level of bias but towards the *upper-right* position instead. The GPT-4o model exhibits the minimal bias among all evaluated VLMs. The grounding bias strongly affects the legend alignment (Figure 12a) where Phi-3.5 performs the worst, GPT-4o has the best performance, while the other 3 models' performance is between them.

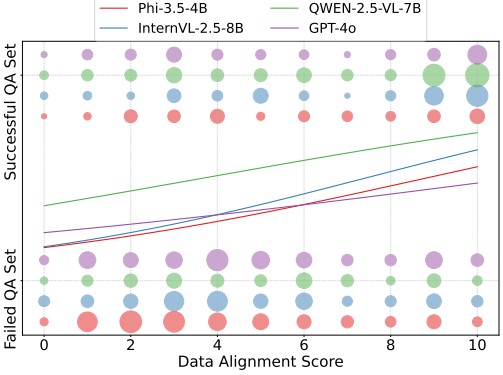

(a) Regression of QA on Data Alignment

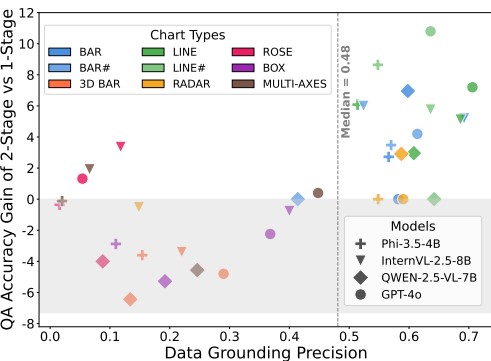

(b) Data Grounding impact on QA

Figure 9: **(a)** shows predicted QA accuracy rising with alignment score, underscoring the dependence of downstream reasoning on accurate fine-grained alignment. **(b)** shows charts with above-median grounding precision consistently yield a positive weighted $\Delta$QA, demonstrating that stronger grounding directly boosts downstream QA. Discussion for both (a) and (b) in Finding 6.

> **Finding 5**
>
> Poor Grounding and Alignment lead to degradation of Downstream QA performance.

Figure 9b demonstrates that precise grounding of visualized-data boosts QA performance. It validates grounding as gateway for extracting structured data from chart for reliable downstream reasoning. Notably, the highest gains occur on simple chart types (bar/line charts and numbered bar/line charts) due to better numeric understanding from chart visual encodings, as discussed in Finding 1. Figure 9a shows steady rise of QA's predicted accuracy with visualized-data alignment demonstrating fine-grained chart understanding's strong association with QA reasoning. These findings position grounding and alignment as essential prerequisites for chart reasoning.

> **Finding 6**
>
> VLM's follow scaling law on alignment tasks.

As shown in Figure 10, we observed a clear scaling law across the various dense alignment subtasks, except for Text-Style Alignment. The deviation arises from the relatively greater complexity of the JSON template in this task, which led to a significantly higher number of failures where InternVL-2.5 produced incorrect JSON formats.

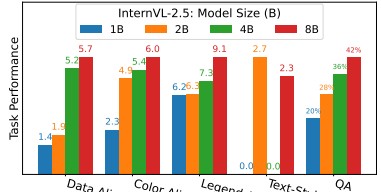

Figure 10: InternVL-2.5 task performance for different model sizes. Other VLM results in Appendix 11.

## 5 CONCLUSION

We introduce ChartAB, the first benchmark for fine-grained chart grounding and multi-chart dense alignment in visionlanguage models (VLMs). Our evaluations across diverse chart types reveal persistent challenges, including perceptual bias, weak attribute understanding, and limited spatial reasoning especially on complex visual representations. Experiments with our novel two-stage pipeline show effectiveness of intermediate grounding in improving dense alignment, and the impact of grounding and alignment accuracy for enhance downstream question answering, establishing these capabilities as essential foundations for robust chart understanding.

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

# A APPENDIX

## A.1 LLM USAGE STATEMENT

LLMs were used in the work as general purpose writing aid (e.g. to polish grammar and phrasing) and to assist with literature search. All substantive research ideation, experiments and analysis has been conducted by the authors.

## A.2 LIMITATIONS

Our work focuses on VLM evaluations and do not assess model fine-tuning. While such approaches might yield stronger results, they diverge from our goal of studying general purpose VLMs for dense level understanding. For dataset construction despite availability of chart datasets with more sophisticated real-world chart examples, we selected the ChartX Xia et al. (2024) dataset because it provides precise chart information in form of csv data and plotting code which is essential for generating precise ground truth values for the evaluation of dense grounding and alignment.

## A.3 DATASET CONSTRUCTION

---

**Algorithm 1:** `ChartAB` dataset curation: *Data Alignment*

---

**Input:** ChartX Xia et al. (2024) dataset $\mathcal{D}$ with CSV table and Python plotting script;
Number of data points to modify $k$;
**Output:** (1) Chart image-pairs, (2) Ground truth - grounding labels: csv-table for each chart,
        alignment labels: describe difference between image-pairs on data points.
**foreach** *chart instance: given csv-table and plotting-script* **do**
    Parse the (ground truth) csv-table $T$ of $x$;
    Identify candidate cells with unique values;
    **if** *fewer than $k$ candidate cells* **then**
        └ **continue**
    Randomly choose $k$ candidate cells and random scaling factors;
    **foreach** *chosen cell $(r_i, c_i)$* **do**
        Compute a modified value by *scaling the column mean*;
        Record original and modified values (for label generation);
    Modify the Python plotting script of $x$ by replacing each original value with its modified
     value (only if a unique match exists);
    Execute the scripts: initial and modified, to generate the image-pair;
    **if** *execution succeeds* **then**
        Add to ground truth label file: grounding label (csv table) and alignment label (JSON:
        cell change, each described by [row name, column name, initial value, modified value]);

---

We used ChartX dataset Xia et al. (2024) as source dataset for our ChartAlignBench curation. ChartX contains plotting-code and csv data-table for the chart with extremely high level of precision thus offering the flexibility for performing finer-level changes along with ground-truth generation capabilities. It contains diverse chart types of varying complexities, and chart data from multiple domains. Hence enabling analysis across charts of varying difficulties.

We utilize *perturbations* for generating fine-grained variations for given chart thus helping build dense-alignment pairs. Chart's plotting-code is perturbed for precise data or attribute changes based on rigorous formatting check using regex-based search and replace, resulting in chart image generation from code execution.

The csv availability and attribute information enable accurate ground-truth generation. Generated pairs for data alignment and attribute alignment include randomly assigned changes, and robustness sets include diverse attribute values for meticulous and unbiased evaluation.

We selected 9 diverse chart types with ability to apply to perform chart data and plot attribute perturbations: (1) *simple charts*: bar chart, bar-numbered chart, line chart, line-numbered chart, (2) *complex charts*: 3D chart, box chart, radar chart, rose chart, multi-axes chart.

---

**Algorithm 2:** Dataset Generation with Attribute Detection and Edits

---

**Input:** Annotated chart dataset $\mathcal{D}$ with drawing scripts and CSV tables.
**Output:** Edited chart images and a JSONL file containing **grounding labels** (detected plot
        attributes) and **alignment labels** (differences between chart pairs).
Initialize output directory and JSONL log file;
**foreach** *chart instance* $x \in \mathcal{D}$ **do**
    Extract Python drawing script $S$ and metadata from $x$;
    `// − Attribute Detection −`
    **Detect Color Lists**: Use regex to locate a unique list of color values. If multiple conflicting
     lists are found, **discard** this chart instance;
    **Detect Legend Position**: Use regex to find the `legend(..., loc=...)` command.
     Discard if a custom (non-standard) specification causes ambiguity;
    **Detect Text Style**: Identify `rcParams` assignments controlling font family, size, and style for
     titles, legends, axis labels, and tick labels;
    Store all detected attributes as the **grounding label** for chart $x$;
    `// − Attribute Modification −`
    **foreach** *attribute type* $a \in \{color, legend position, text style\}$ **do**
        **switch** $a$ **do**
            **case** *color* **do**
                Randomly replace some or all colors in the detected list with new randomly
                 generated colors;
            **case** *legend position* **do**
                Replace the detected `loc` value with a randomly chosen valid location;
            **case** *text style* **do**
                Randomly vary selected `rcParams` such as font family or size;
    Generate modified script $S_a$ with updated attribute values;
    Update figuresave path in $S_a$ and execute it to render the altered chart image;
    **if** *rendering succeeds* **then**
        Create a JSON record containing:
            • Image name and chart type,
            • Detected (original) attribute values,
            • Modified attribute values.
        Store these as the **alignment label**, describing the attribute difference between the
         original and edited chart;
        Append the record to the JSONL log file;
Report overall success statistics (processed charts vs. total);

---

## A.4 DENSE ALIGNMENT TASKS

**Data Alignment.** The task evaluates data alignment in image pairs, i.e., difference in values of cells in the data table, which is visualized by the charts. The finer-level cell changes involve performing (1) *1-cell change*, (2) *2-cell change*, (3) *3-cell change* between the chart images. The task aims to analyze the model's ability to perceive change in visual encoding property (e.g., position, shape, size) in the chart image, and ability to map it to the specific cell, i.e., row & column headers in data-table modality, along with measuring the cell change utilizing the visual components of the image describing scale and values.

**Attribute Alignment.** The task evaluates attribute alignment in image pairs, i.e., difference in values of visual attributes which are part of the chart design. We assess the capability through three alignment tasks:- (1) *color alignment*, (2) *legend alignment*, (3) *text-style alignment*. The plot-alignment task aims to analyze model's ability to perceive finer-level visual design changes.

1. *Color Alignment* evaluates alignment of encoding colors, i.e. difference in colors of visual encodings representing chart data: bars in bar chart, lines in line chart, segments/spokes in rose chart etc.

2. *Legend Alignment* evaluates alignment of legend, i.e. difference in position of legend in the charts.

3. *Text-Style Alignment* evaluates alignment of text characteristics namely (1) *size*, (2) *weight* i.e. degree of boldness (3) *font-family* i.e. style of font applied. The text in chart corresponds to following chart sections: title, legend, axes-labels, axes-ticks.

Overall plot-alignment task aims to analyze model's ability to perceive change in visual design characteristics (e.g. visual encodings, axes, labels, legends) in the chart image, and semantic understanding to map it to specific attribute. And ability to precisely predict the attribute value from representation and component structure of the chart.

**Robustness.** The task evaluates the robustness of data alignment against variation of plot attributes, namely, colors, legend, and text style. To evaluate how the data alignment ability of VLMs varies when changing each visualization attribute or plot design, we curate a dataset such that each instance is composed of several pairs of charts, all based on variations of the same source chart. All the pairs share the same pairwise difference in the data values but differ in certain plot visualization attributes.

Robust models are expected to generate consistent responses despite varying attributes, while variation in model responses can quantify sensitivity to the changes in plot designs. Hence, the study provides robustness metrics of data alignment against plot attributes.

## A.5 A Two-Stage Evaluation Pipeline: Details

We utilize natural-language based instructions for zero-shot inference to enable simple execution with minimal task specific nuances for strong generalization across various models.

VLM outputs follow *JSON based formatting* due to precise nature of the key-value structure which is essential for element specific information serialization for finer-analysis, along with flexibility for variations in completion of grounding and fine grained analysis. The alignment JSON contains finer level attributes for which the charts differ, and the values for corresponding attribute in the two charts. E.g. for data alignment (as shown in Fig. 4) the finer level attributes changed between the charts i.e. cells are identified by their row & column header, along with its values in the chart pairs, i.e. value in chart 1 & value in chart 2 respectively. Evaluation of attribute alignment tasks follow the same pipeline, as illustrated in Figure 15 for color alignment, Figure 16 for text-style alignment, Figure 12a for legend alignment.

## A.6 Evaluation metric: Alignment

Alignment evaluation is done by calculating similarity of VLM's evaluation response JSON vis-a-vis the ground-truth anchor. The JSON encompasses finer-level *constituents* (e.g. bars of bar chart with color-difference in color-alignment task) which differ between the chart-pairs along with their specific value, and are evaluated for their correctness.

### A.6.1 Attribute Alignment

For attribute alignment score, the accuracy for each $N$ constituent is calculated for the chart-pair (chart-1 & chart-2), and averaged for all constituents to get the score. The Accuracy $A_i$ is calculated based on the alignment task, contrasting the evaluation response value with the ground-truth value.

$$\text{Score} = 10 \cdot \left( \frac{1}{N} \sum_{i=1}^{N} \mathcal{A}_i \left( chart_1 \right) + \mathcal{A}_i \left( chart_2 \right) \right) \tag{1}$$

*Legend Accuracy*: The legend position accuracy using the manhattan distance, the position associated with the 3 by 3 grid:

$$\mathcal{A}^{\text{legend}} = 1 - \frac{1}{5}\text{Manhattan}(\text{position}, \hat{\text{position}}) \tag{2}$$

*Color Accuracy*: The color accuracy is calculated using L1 distance:

$$\mathcal{A}^{\text{color}} = 1 - \frac{1}{3} \sum_{i \in \{R,G,B\}} \frac{|intensity_i - \hat{intensity}_i|}{255} \tag{3}$$

*Text Accuracy*: The text alignment accuracy is calculated by correctness of size, weight, fontfamily respectively.

$$\mathcal{A}^{\text{text style}} =$$

$$\frac{1}{4} \sum_{i \in \{\text{title,legend,ticks,labels}\}} (0.4 \cdot \mathbb{1}[\mathbf{size}_i = \hat{\mathbf{size}}_i] \tag{4}$$

$$+ 0.3 \cdot \mathbb{1}[\mathbf{weight}_i = \hat{\mathbf{weight}}_i]$$

$$+ 0.3 \cdot \mathbb{1}[\mathbf{fontfamily}_i = \hat{\mathbf{fontfamily}}_i])$$

### A.6.2 DATA ALIGNMENT & ROBUSTNESS

Data Alignment score calculation follows the JSON correctness discussed in evaluation metrics section. However data alignment accuracy is calculated for the combined image-pair, unlike individual image in attribute. As for data alignment we also evaluate the correctness of the finer-level constituent's key (i.e. identification) which are the cell's row & column name whereas in attribute alignment only constituent's value is evaluated. Data alignment scores are also averaged for all chart-pairs in a chart-type. For $N$ being the number of cell-change between the image-pairs, data alignment score is defined as:

$$\text{Score} = 10 \cdot \left( \frac{1}{N} \sum_{i=1}^{N} \mathcal{A}_i^{\text{cell}} (chart - pair) \right) \tag{5}$$

The cell accuracy $A^{\text{cell}}$ is determined by the cell' value accuracy (for each chart), and the evaluation response's row & column similarity (for chart-pair).

$$\mathcal{A}^{\text{cell}} = 0.3 \cdot \text{Sim}^{\text{row}} + 0.3 \cdot \text{Sim}^{\text{col}} \tag{6}$$

$$+ 0.2 \cdot A_{\text{chart-1}} + 0.2 \cdot \text{Val}_{\text{chart-2}}$$

The row and column name correctness is evaluated using Levenshtein distance based string comparison:

$$Sim^i = Levenshtein(i, i^) \tag{7}$$

The cell-value accuracy (for a chart) is evaluated using the percentage value difference:

$$\text{Val}_i = max \left( 1 - \left( \frac{|\text{cell\_val} - \hat{\text{cell\_val}}|}{\text{cell\_value}} \right), 0 \right) \tag{8}$$

*Robustness*: Robustness of data alignment over variation in attribute aims to evaluate model's ability to maintain consistent alignment over changing attributes. The data alignment score is utilized for developing the robustness evaluation metric. For robustness, each chart has set of 5 data alignment pairs with identical data alignment but variation in attribute values. We define $\mu(\text{set})$ and $\sigma(\text{set})$ as the mean and standard-deviation respectively of the 5 image-pairs in the robustness set for a chart.

$\sigma(\text{set})$: It represents the deviation of 5 chart pairs. A high value indicates of large difference between the data alignment scores of the chart-pairs hence low robustness.

We define the Robustness metric as reciprocal of mean of $\sigma(\text{set})$ for all the charts, for particular configuration: i.e. cell-change $c$, and the altered attribute $p$.

$$R(c,p) = \frac{1}{\frac{1}{N_{c,p}} \sum_{\substack{\text{cell-change}=c \\ \text{attribute}=p}} \sigma(\text{robustness set})} \tag{9}$$

### A.7 ADDITIONAL EXPERIMENTAL DETAILS

### A.7.1 VLM SELECTION

We evaluate a diverse suite of *open-source VLMs* from following families: Phi-3.5 vision-instruct Abdin et al. (2024), InternVL-2.5 (8B) Chen et al. (2024), LLaVA-1.6 Mistral (7B) Liu et al. (2023a),

QWEN-2.5 VL (8B) Bai et al. (2025). These models constitute among most widely used VLMs, and have a long timeline of continuous evolution with each released version. The set encompasses the top-performed VLMs in various chart benchmarks (CharXiv Wang et al. (2024b), ChartQAPro Masry et al. (2025), SCI-CQA Li & Tajbakhsh (2023), MultiChartQA Zhu et al. (2024), discussed in 2).

Our choice of *proprietary VLM* is based on CharXiv Wang et al. (2024b) leaderboard as its tasks/questions require dense-level grounding. For example, CharXiv tasks need to identify axes ticks by positions and their value enumerartion, grid-lines count and intersections, integral (area comparison of regions) and slope (rate of increase/decrease) in line charts. And GPT-4o Hurst et al. (2024) is the best performing proprietary in the CharXiv paper.

Among *chart-specialized VLMs*, we evaluate TinyChart Zhang et al. (2024b) & ChartGemma Masry et al. (2024) models. However, due to their task-specific training (discussed in 2), these models show collapse of instruction following capabilities and fail to output required JSON format needed for evaluation. Below are a few examples of the outputs.

JSON output: Data alignment (1 cell) by ChartGemma and TinyChart models using 1-stage stitched-charts (i.e chart pair stacked as single image) evaluation.

```
REQUIRED FORMAT (specified in prompt instructions):-
{"row name": <row name of the cell>, "column name": <column name of the cell>,
 "value in chart 1": <value in first chart of the pair>, "value in chart 2": <value in second chart of the pa
```

```
EXAMPLE:-
{"row name": "Production A (million units)", "column name": "2021",
 "value in chart 1": 35, "value in chart 2": 30}
```

```
CHARTGEMMA OUTPUT (abnormal valued JSON which is inconsistent with required format):-
{"row name": "sample row", "column name": "sample column",
 "value in chart 1": Infinity, "value in chart 2": Infinity}
```

```
TINYCHART OUTPUT (abnormal list instead of JSON):-
["Production A (million units)", "Production B (million units)",
 "Production C (million units)" ..... "Production Z (million units)"]
```

### A.7.2 ABLATIONS

| Type | Approach | Bar | Bar # | 3D Bar | Line | Line # | Radar | Rose | Box | Multi-Axes |
|------|----------|-----|-------|--------|------|--------|-------|------|-----|------------|
| 1-stage | Multi-chart | 4.8 | 7.4 | 4.7 | 3.3 | 4.7 | 4.9 | 3.1 | 3.2 | 3.3 |
|  | Stitched-chart | 5.0 | 4.8 | 3.0 | 4.5 | 3.5 | 3.0 | 2.7 | 2.8 | 3.2 |
| 2-stage | Ours | 6.5 | 8.3 | 4.1 | 6.1 | 6.3 | 3.8 | 3.4 | 2.9 | 3.5 |

Table 1: **Ablation study of 1-stage vs. 2-stage evaluations** on data alignment (one cell change) task. Mean scores across nine chart types show that our 2-stage evaluation reflects VLMs' greatest potential on chart alignment.

We performed ablation experiments to vigorously compare differing approaches to our 2-stage approach.

The ablation experiments aimed to thoroughly compare single-stage based alignment approaches for performing multi-image reasoning vis-a-vis our two-stage approach. The ablation techniques:-

(1) *stitched-charts* inference: The chart-pair images are vertically concatenated resulting in a single image of stitched chart-pairs which undergo single-stage inference.

(2) *multi-image* inference: The VLM inputs multiple images, and contextualizes output based on the input images with aim of better understanding across of finer-level alignment in multi-image reasoning.

The ablation experiments analyzed Phi-3.5 model's performance on data alignment task. As shown in table. 1, the single-stage approach fared poorly compared to out two-stage approach reaffirming the

two-stage approach. Multi-image inference showed the weakest performance. Despite increasing training efforts towards improved VLM training, the models still face issues in reasoning ability on fine-grained tasks. Stitched-charts approach showed better results than multi-image, however they too underperformed vis-a-vis our two-stage approach. The comparatively stronger image self-attention capabilities seem to augment multi-image by utilzing the stitched connection. However the better prevailing capabilities of two-stage approach capture the gain of grounding generation. The VLM's multi-modal understanding though improving still suffers from finer-level nuances missed by information loss in image-encoding and cross-attention mechanisms.

## A.8 ADDITIONAL FINDING & INSIGHTS

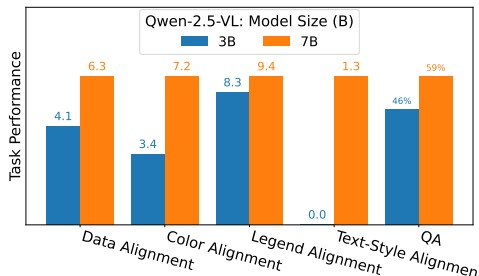 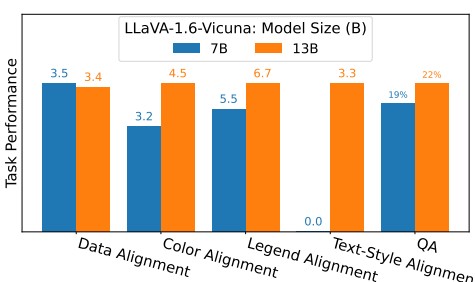

Figure 11: Task performances for different sizes of Qwen-2.5-VL and LlaVa-Vicuna-1.6.

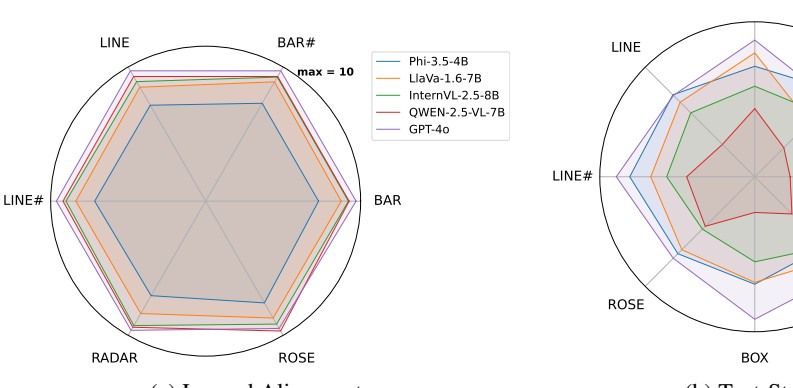

(a) Legend Alignment  (b) Text-Style Alignment

Figure 12: (a) **Legend alignment** of legend positions. Phi-3.5 performs the worst while GPT-4o is best. Related discussion in Finding 1&2. (b) **Text-style alignment** (size, weight, font). Worst: QWEN-2.5-VL, Best: GPT-4o. Discussion in Finding 1&4.

---

**Finding 7**

VLMs' data grounding and alignment are more robust to color variations than changes in legend positions and text styles.

---

Fig. 13 shows that robustness is the worst under text-style variations and the best under color variations. In the visualizations of data, colors are used to discretize, categorize, and measure chart constituents. As long as their colors are distinguishable, color variations will not affect the data grounding. In contrast, the text styles and legends provide critical information about the data via ticks, labels, and legend items. Moreover, changing legend position may lead to position changes and occlusion of other chart elements. Hence, their variations have a greater impact on the data grounding/alignment performance.

---

**Finding 8**

VLMs' spatial understanding capability affects several important chart understanding skills.

---

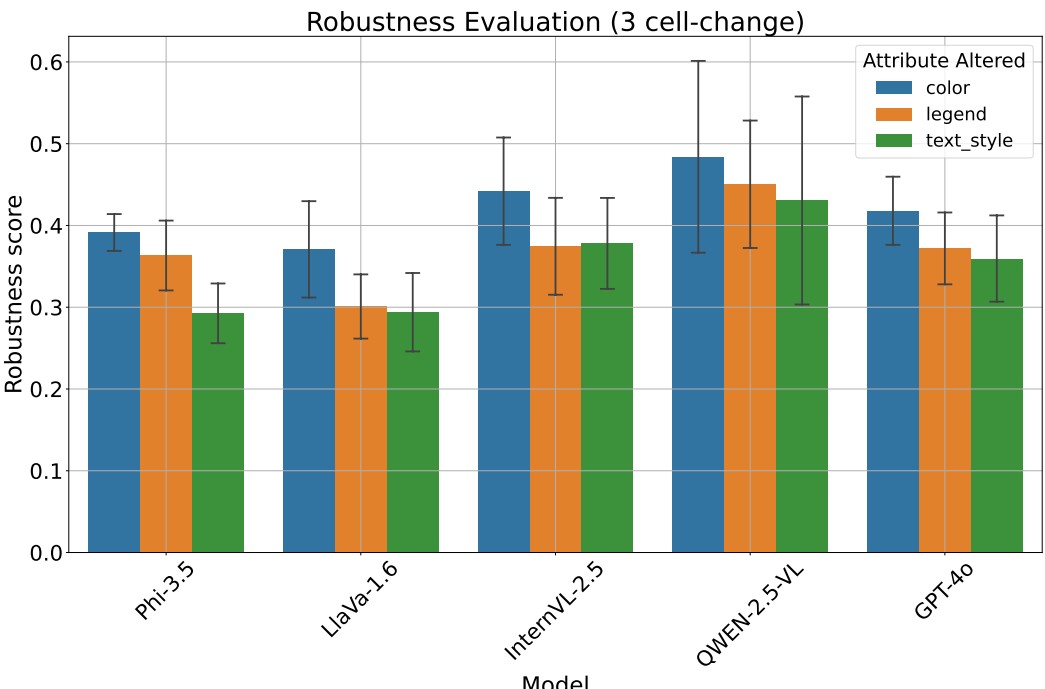

Figure 13: **VLMs' Robustness of data alignment (3-cell change) to variations in color, legend, and text-style.** VLMs show better robustness to color changes than text-style changes. QWEN-2.5-VL outperforms the other four VLMs on robustness. More discussion can be found below Finding 6.

Chart understanding usually requires an accurate mapping between spatial relationships and the corresponding numerical values to be visualized.

- *Depth understanding*: Despite the high-level similarity between 3D bar charts and (2D) bar charts, as shown in Fig 5, the data alignment performance is much poorer on 3D bar charts due to the lack of depth understanding, which affects the measurement of scales and values along axes in the 3D space.

- *Text vs non-text cues*: Rose charts are extended from bar charts by allowing more polar coordinates with scale differences in radial forms. However, Fig. 14b reveals a great difference between the two on data alignment performance. This is due to fewer text cues (e.g., axes ticks) in rose charts, where non-text cues such as grid lines cannot be fully leveraged.

- *Better performance on numbered charts*: numbered bar and line charts explicitly place the data values in the charts, hence facilitating VLMs to extract the data easily without precise measurements of the visual elements. Hence, as shown in Fig. 5, numbered bar/line charts usually enjoy better performance.

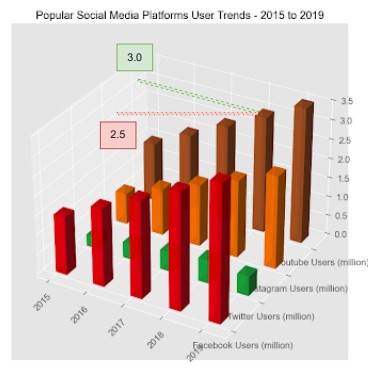

(a) Depth estimation in 3D bar charts

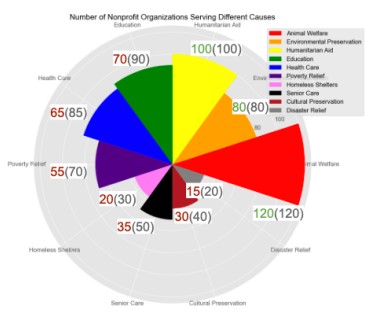

(b) Text vs. non-text cues for value scaling in rose charts.

Figure 14: **VLMs' spatial understanding is poor on complex charts.** More discussion is provided below Finding 7.

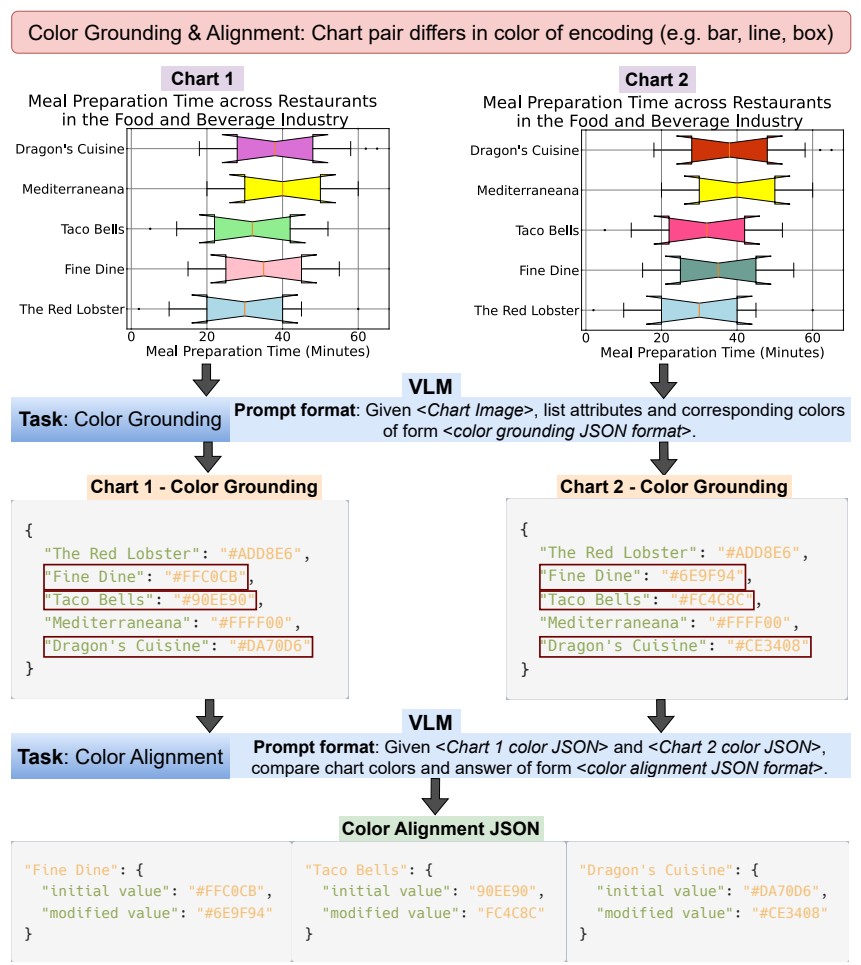

Figure 15: **COLOR ALIGNMENT task ChartAB.**

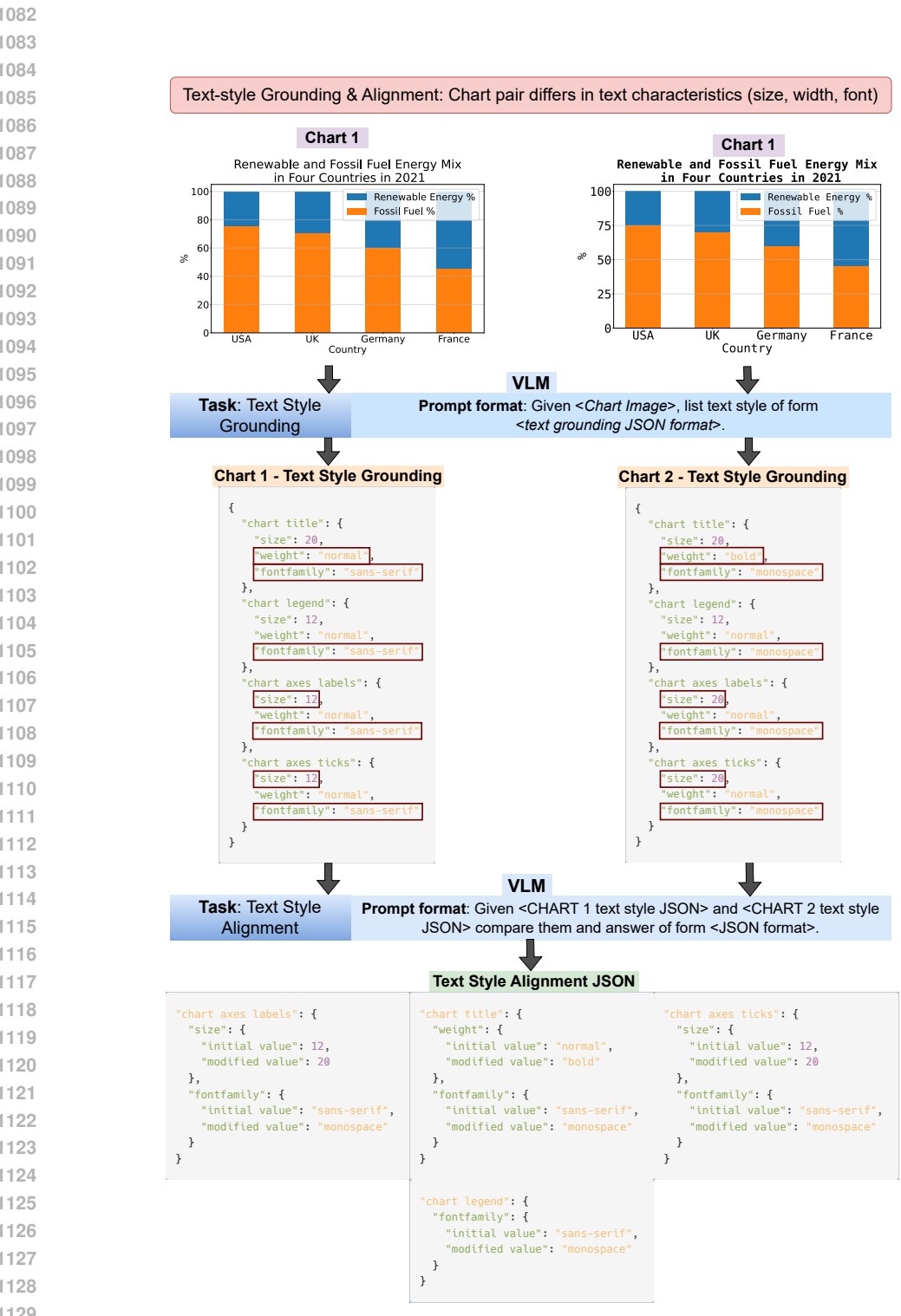

Figure 16: **TEXT STYLE ALIGNMENT task `ChartAB`.**

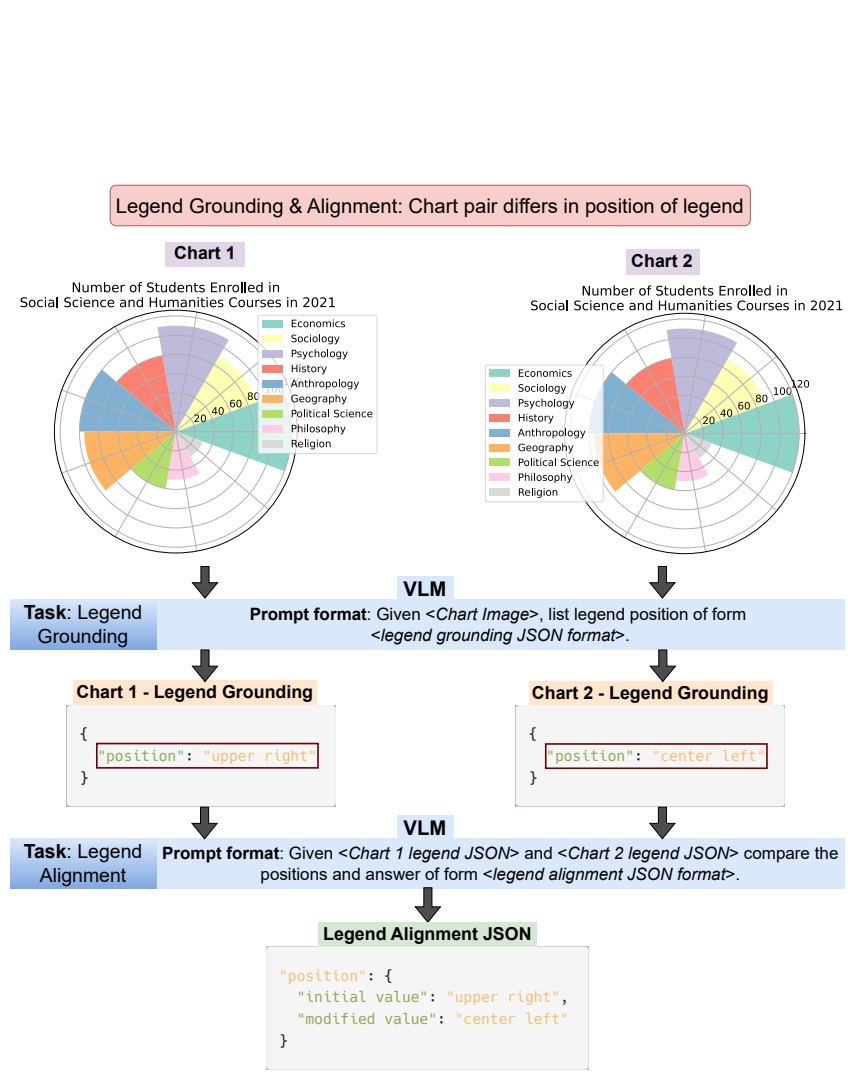

Figure 17: **LEGEND ALIGNMENT task `ChartAB`.**

