# OpenReview forum: "ChartAlignBench: A Benchmark for Chart Grounding & Dense Alignment"
_ICLR.cc/2026/Conference — ICLR 2026 Conference Withdrawn Submission_

### Official Review · Reviewer_Xkh1 · 2025-10-21

**Soundness:** 3
**Presentation:** 3
**Contribution:** 1
**Rating:** 2
**Confidence:** 4

**Summary:**

This paper introduces the ChartAlign Benchmark (ChartAB), a benchmark designed to evaluate the fine-grained perceptual abilities of Vision-Language Models (VLMs) in chart understanding. The authors argue that existing benchmarks focus on simple QA, failing to assess detailed comprehension. ChartAB addresses this gap by proposing two core tasks: (1) chart grounding, which requires VLMs to extract structured information, such as tabular data and visual attributes (e.g., color, text style, legend position), into a JSON template, and (2) dense alignment, which uses a two-stage inference workflow to test a model's ability to compare and identify subtle differences between two similar charts. By evaluating several recent VLMs, the authors reveal significant weaknesses, biases, and hallucinations in current models' abilities to perceive complex chart structures, demonstrating that these foundational skills are critical for robust downstream reasoning.

**Strengths:**

The authors have conducted abundant experiments.

**Weaknesses:**

- The paper's core terms, "grounding" and "alignment," are used in a non-standard and confusing way. In V&L, "grounding" typically implies localizing language to specific spatial regions (e.g., bounding boxes). This paper redefines it as extracting information into a structured textual (JSON/CSV) format. This is more accurately described as data extraction or structured representation. Similarly, the term “alignment” usually refers to mapping representations between modalities (e.g., vision and language), or making models conform to certain preferences. The paper uses it to mean a simple comparison between two already extracted textual representations to find differences. This terminological choice overstates the novelty and creates confusion.

- The practical utility of the "Attribute Grounding & Alignment" task is not well-motivated. The paper claims comparing attributes is an "essential skill", but it's unclear why a user would need a VLM to identify that one chart uses a "bold" font and another uses a "normal" font, or that one legend is on the left and another is on the right. The other two tasks also do not seem very practical but slightly more useful than "Attribute Grounding & Alignment".

- ChartAB builds heavily on the existing ChartX dataset, with perturbations applied to create pairs. While this adds pairs for comparison, it doesn't sufficiently differentiate from prior chart benchmarks (e.g., ChartQA, CharXiv, MultiChartQA), which already cover QA, multi-hop reasoning, and multi-chart tasks. The focus on "dense" extraction feels incremental rather than groundbreaking, especially since similar grounding approaches (e.g., DePlot for image-to-CSV) are cited but not substantially advanced.


- While the paper identifies VLM weaknesses like hallucinations, biases, and perceptual inaccuracies, it stops at analysis without proposing concrete enhancements (e.g., fine-tuning strategies or architectural changes). This makes the benchmark feel more diagnostic than constructive, limiting its impact on advancing VLMs for chart tasks.

**Questions:**

What is the practical, real-world scenario for the "Attribute Alignment" task?

---

### Official Review · Reviewer_vTfu · 2025-10-31

**Soundness:** 2
**Presentation:** 2
**Contribution:** 2
**Rating:** 2
**Confidence:** 3

**Summary:**

This paper proposes ChartAlignBench, a benchmark on chart dense grounding and alignment. The authors define dense grounding and alignment as identifying data variations and chart element differences such as colors, styles, and legends. The dataset contains 9k pairs of chart images. The visualizations are generated by perturbing certain code content with an LLM. The authors also contribute a two-stage evaluation pipeline and show weaknesses in some VLMs.

**Strengths:**

+ The two-stage evaluation pipeline makes the evaluation more rigorous, with ablations supporting this.
+ The dataset size, 9k, is sufficiently large.
+ The presentation of the paper is overall easy to follow.

**Weaknesses:**

- My central concern is the utility of the benchmark. I don't agree with the authors that "real-world use cases often require comparing similar charts to detect subtle differences among the charts". While it might be true that sometimes one needs to compare a series of visualizations to check differences in data, I cannot imagine tasks such as comparing colors of chart elements or fonts of text being at all common, but these questions take up a significant portion of the benchmark.
- The models evaluated are quite old. The best model, GPT-4o, is more than one year old at this point. No reasoning models are evaluated. Reasoning models have made tremendous progress in chart grounding, so I suspect the numbers will be a lot higher.
- Most questions assess some form of perception, and a lot of it is meaningless or unanswerable. For things like identifying the font weight, you simply cannot do this. If you fix the font weight and make the whole chart image bigger, the text is going to be bigger. Overall I just don't find tasks like "comparing the hex color of chart elements" to be an interesting exercise.
-The visualizations showing model performance are pretty difficult to parse. Presenting them in a table format would be better.
- Given that the tasks cover a pretty fixed set of visualization design space (e.g., color, legend placement, fonts), I would expect finetuning to be really helpful, which the authors did not perform.

**Questions:**

- How does the two-stage pipeline work for non-data-alignment questions?
- What does Figure 9a mean? What does the size encoding channel encode?

---

### Official Review · Reviewer_KT8b · 2025-10-31

**Soundness:** 2
**Presentation:** 1
**Contribution:** 2
**Rating:** 2
**Confidence:** 4

**Summary:**

This paper introduces ChartAB, a novel framework for evaluating/probing VLMs (VLMs) on chart grounding (extracting structured data and attributes from individual charts) and "dense alignment" (identifying fine-grained differences between chart pairs) tasks. ChartAB derives from ChartX, retaining good diversity of domains and chart types over 9k examples, but is structured around spreadsheet extraction; attribute extraction for visual encoding colors, legend location, and text style; and contrasting pairs or charts partially differing in data or attribute. Authors argue that VLMs must be evaluated by sequentially performing grounding and then dense alignment, with a JSON-like interface between the two, and proceed to analyze 4 open-source VLMs + GPT-4o through the lens of this benchmark.

**Strengths:**

1. Drawing relationships between charts ("alignment") on a panel or dashboard is a relevant motivation, and it is indeed a relative blind spot in the data resources space compared to individual charts.

2. The number of examples ("9k pairs"---although an exact number would be good) is good, as well as the diversity of chart types. The separation between data grounding and visual encoding understanding is also meaningful, with the "alignment" part being the main novelty and contribution.

**Weaknesses:**

The paper presentation is confusing and excessively repetitive in some parts. Importantly:

1. Being a benchmark, it is crucial to report the exact performance measurements of the baselines tested. Without them, the current choice of charts through Section 4 feels unclear and inappropriate. For example, on Figure 5, why do we only see a radar chart with nearly overlapping results exclusively for the one cell configuration (whereas the benchmark included two and three cells)? A well organized table is crucial for communicating exactly how far the current set of tested models go on the benchmark's task(s), along with a clear definition of the performance measure(s) close to the table. This feedback is applicable throughout Section 4.

2. Lines 222-223 refer to Appendix A.7.2 implying that the 2-stage pipeline is unequivocally better, but on Table 1 the 1-stage multi-chart configuration is superior on 3 out of 9 chart types (3D Bar, Radar, and Box). Table 1 should be better organized (see #1 above---specifically, performance definition should be clearly linked, and whether higher/lower is better) and these mixed results should be more transparently discussed.

3. VisText [1] and InsightBench [2] should be cited, as much of the value from drawing relationships between charts is to be able to draw insights and understand trends, which are the topic of these other data resources.

4. Examples of writing that can be improved include:

i. Lines 168-172 are repetitive w.r.t. lines 162-167.

ii. Lines 312-319 feel particularly confusing.

iii. Sentences in lines 91-92 and 338-339 could skip the break and be merged.

[1] Tang, Benny, Angie Boggust, and Arvind Satyanarayan. "VisText: A Benchmark for Semantically Rich Chart Captioning." In Proceedings of the 61st Annual Meeting of the Association for Computational Linguistics (Volume 1: Long Papers), pp. 7268-7298. 2023.

[2] Sahu, Gaurav, Abhay Puri, Juan A. Rodriguez, Amirhossein Abaskohi, Mohammad Chegini, Alexandre Drouin, Perouz Taslakian et al. "InsightBench: Evaluating Business Analytics Agents Through Multi-Step Insight Generation." In The Thirteenth International Conference on Learning Representations.

**Questions:**

As described in weakness #1: For each task in the benchmark, could the authors include a table with the exact performance measurements of the baselines tested?

As described in weakness #2: Could the authors provide more details on the mixed results reported on Table 1?

---

### Note · Authors · 2025-11-14

I have read and agree with the venue's withdrawal policy on behalf of myself and my co-authors.